# A scoping review of the impact of organisational factors on providers and related interventions in LMICs: Implications for respectful maternity care

**Bhavya Reddy**[1,2]*, **Sophia Thomas**[1], **Baneen Karachiwala**[1], **Ravi Sadhu**[3], **Aditi Iyer**[1], **Gita Sen**[1], **Hedieh Mehrtash**[4], **Özge Tunçalp**[4]

**1** Ramalingaswami Centre on Equity and Social Determinants of Health, Public Health Foundation of India, Bangalore, India, **2** School of Population and Public Health, University of British Columbia, Vancouver, Canada, **3** T.H. Chan School of Public Health, Harvard University, Boston, Massachusetts, United States of America, **4** Department of Sexual and Reproductive Health and Research, World Health Organization, Geneva, Switzerland

* bhavya@reddyb.com

**Data Availability Statement:** All relevant data are in the paper and its Supporting Information files.

**Funding:** This work was funded by UNDP/UNFPA/UNICEF/WHO/World Bank Special Programme of

## Abstract

We have limited understanding of the organisational issues at the health facility-level that impact providers and care as it relates to mistreatment in childbirth, especially in low- and middle-income countries (LMICs). By extension, it is not clear what types of facility-level organisational changes or changes in working environments in LMICs could support and enable respectful maternity care (RMC). While there has been relatively more attention to health system pressures related to shortages of staff and other resources as key barriers, other organisational challenges may be less explored in the context of RMC. This scoping review aims to consolidate evidence to address these gaps. We searched literature published in English between 2000–2021 within Scopus, PubMed, Google Scholar and ScienceDirect databases. Study selection was two-fold. Maternal health articles articulating an organisational issue at the facility- level and impact on providers and/or care in an LMIC setting were included. We also searched for literature on interventions but due to the limited number of related intervention studies in maternity care specifically, we expanded intervention study criteria to include all medical disciplines. Organisational issues captured from the non-intervention, maternal health studies, and solutions offered by intervention studies across disciplines were organised thematically and to establish linkages between problems and solutions. Of 5677 hits, 54 articles were included: 41 non-intervention maternal health-care studies and 13 intervention studies across all medical disciplines. Key organisational challenges relate to high workload, unbalanced division of work, lack of professional autonomy, low pay, inadequate training, poor feedback and supervision, and workplace violence, and these were differentially influenced by resource shortages. Interventions that respond to these challenges focus on leadership, supportive supervision, peer support, mitigating workplace violence, and planning for shortages. While many of these issues were worsened by resource shortages, medical and professional hierarchies also strongly underpinned a number of organisational problems. Frontline providers, particularly midwives and nurses,

Research, Development and Research Training in Human Reproduction (HRP), a cosponsored program executed by the World Health Organization (WHO). The funders had no role in study design, data collection and analysis, decision to publish, or preparation of the manuscript.

**Competing interests:** The authors have declared that no competing interests exist.

suffer disproportionately and need greater attention. Transforming institutional leadership and approaches to supervision may be particularly useful to tackle existing power hierarchies that could in turn support a culture of respectful care.

## Background

Improving maternal health has remained a global health and development priority over several decades, reflected in the Millennium Development Goals and the current Sustainable Development Goals. Among key strategies, governments across low- and middle-income countries (LMICs) have focused on increasing the use of institutional delivery as a means to reduce maternal mortality [1]. As a result, more women now give birth in facilities than ever before. But increased institutional childbirth has exposed serious problems and inequities in both clinical quality and women's experiences of care [1,2]. We have significant evidence that women, especially those who are disadvantaged and marginalised, experience various forms of mistreatment when accessing maternal care in facilities and especially during childbirth—from being shouted at and restrained, to being subjected to unnecessary and invasive obstetric practices, often in the absence of communication and consent [2–4]. Mistreatment in maternal care has also been extensively reported in high-income countries. Abuse, callousness, discriminatory attitudes from providers, patients not being given adequate information on procedures, and being excluded from decision-making around care, among other issues, have been documented [5–7]. Growing evidence on these issues has galvanised a global movement to recognise women's rights in childbirth (and through the care continuum), focused research to understand mistreatment, and a policy agenda to promote what is widely known as Respectful Maternity Care (RMC) [8,9].

In previous work we have argued that mistreatment arises from intersecting social and economic inequalities, and the institutional structures and processes that frame the practice of obstetric care [10]. A systematic investigation and critical interpretative synthesis of drivers from this Collection supports and advances these arguments. Schaff et. al. frame power as central to our understanding of mistreatment, identifying organisational power dynamics and the pressure to achieve health system performance goals among forces that can work against women-centred, respectful care [11].

An extensive body of literature from high-income countries (HICs), rooted in organisational behaviour, organisational psychology, among other disciples, has explored aspects of the organisational and work environment that impact providers across different branches of medicine. These factors span working relationships, management practices, leadership style [12,13], professional autonomy [14], structural issues such as physical environments and shift length and timings [15,16], as well as issues related to workforce shortages [17]. The research on their impact on providers cluster around: psychological well-being (expressed through depression, burnout, stress and anxiety etc.) [18]; motivation (job satisfaction, organisational commitment, absenteeism, intention to leave); performance (work engagement, resourcefulness, involvement), as well as factors that translate directly to care provision (attentiveness, empathy, missed care, medical errors and other elements of patient safety). The concepts of caring and compassion in relation to these working conditions have been examined in-depth in the nursing literature [19,20]. We also know that maternal healthcare providers may be particularly at risk of burnout due to stressors specific to intrapartum care such as the long length of labour, the risk of complications at any time, and balancing the dual responsibility of maternal and foetal outcomes [21,22].

In comparison, studies on maternal care in LMICs have more clearly pointed to health system pressures related to understaffing, high patient loads, and other resource shortages as triggers for poor provider behaviour [3,23–25]. But even in LMIC contexts where shortages are acute, challenges in working environments and provider responses to them may not stem from resource constraints alone. How work is divided, supervised, and rewarded are important factors for provider motivation and performance across all regions [26–29], which may be shaped by formal institutional norms as well as less tangible factors related to workplace culture [30,31]. Studies in LMIC settings have cited hierarchical organisational structures, inadequate support for providers, and a culture of blame [28–31] among factors that contribute to provider burnout and low morale. Working conditions that impact provider morale and job satisfaction may be associated with or precede disrespectful and abusive behaviour in maternity care [23,32], as they have been clearly associated with poor patient satisfaction more broadly [33]. The research on these issues in LMICs, however, have not been consolidated to highlight which organisational factors matter for respectful maternity care and why, and the extent to which they are connected to shortages.

The gap between HIC and LMIC literature also extends to knowledge on how organisational interventions can improve provider behaviour and care. In HIC settings, the implications of organisational problems at the facility level, including working conditions, have been well recognised, prompting considerable intervention research in the area. This extends to intervention research with a focus on "caring for carers", on improving support for providers within and beyond healthcare [34]. Studies from HICs that aim to improve provider wellbeing and performance have been targeted at the organisational level through measures like reducing shift lengths and enabling rest, increasing professional support, and at the individual level through increasing coping mechanisms–such as mindfulness and meditation workshops and resiliency training [35–37]. Such intervention research is more elusive in LMICs where much of the focus has been on large-scale health systems strengthening programmes, and policy efforts to increase the human resource workforce and their retention, strengthen clinical training, upgrade infrastructure, bolster and streamline financing, and improve health system governance [38,39]. Our understanding based on preliminary searches is that, barring a few promising initiatives [26,40], research exploring and addressing organisational factors at the facility level that impact how care is provided, is relatively limited in LMICs and inadequately explored in the context of enabling RMC. Further, while pressures related to organisational issues may co-exist with various types of shortages in LMIC settings, we know less about how shortages interact with other organisational conditions to shape how care is provided and experienced.

## Aims

This scoping review aimed to identify key organisational factors at the facility-level that impact provider behaviour and the experience of maternity care in LMIC settings. It also seeks to unpack the ways in which they are influenced by or are a response to resource constraints. Our second aim is to identify organisational changes or changes in working environments that have a bearing on provider behaviour or aspects of care provision that are relevant to RMC.

The review aims to address the following questions:

1. What are key organisational factors that impinge on provider behaviour in healthcare settings in LMICs?

   - How are they influenced by or a response to resource constraints?

2. What changes to organisational structure and processes within resource poor contexts can improve provider behaviour toward patients and/or the experience of care?

- What lessons can be drawn for interventions aiming to enable Respectful Maternity Care in LMICs?

## Methodology

We draw on two constructs from the field of Health Policy and Systems Research (HPSR) to situate the area of inquiry and define its broad boundary. Health systems may be viewed as having multiple levels of operation: macro, meso, and micro [41]. Our focus is meso-level factors, that is facility level and institutional environments, which includes the management of health workers [42]. Our understanding is that facility-environments have been less studied in LMIC contexts in comparison to health system wide conditions and challenges. Gilson and colleagues [42] also propose a framework of the 'terrain of HPSR', which describes health systems as comprising hardware (structure, organisation, technology, and resourcing) and software (values, norms, actors and relationships). We explore components of both hardware and software at the facility-level, specifically:

- Division of labour: how work is divided among cadres and organised across clinical, administrative, and other duties;

- Supervision and monitoring;

- Working conditions, including but not limited to resource constraints; and

- Workplace culture.

### Inclusion criteria

An exploratory phase of screening literature was conducted to refine the inclusion criteria. In this phase, we kept the search open to all regions and branches of medicine. We found that an overwhelming volume of empirical inquiry and intervention in the area of organisational factors in healthcare (and its impact on providers) was from high-income country settings, and a considerable subset situated in maternal healthcare in HICs. It was comparatively less explored for maternal healthcare in LMIC contexts. As the area of interest for this review was not only organisational factors at the facility-level, but also the relationship to resource constraints, we found it important to narrow the scope of the review to LMIC contexts where shortages can be acute. However, in terms of interventions studies, the focus on maternal health yielded too few articles for useful analysis, therefore, intervention study criteria remained open to all medical disciplines in LMICs.

Peer-reviewed empirical journal papers from LMICs with a focus on maternal health, and intervention studies from all disciplines, were included if they were: published between the period of 2000–2021, written in English, described organisational factors that impinge on provider behaviour and/or aspects of care provision relevant to RMC. Quantitative, qualitative, and mixed-method studies were included in order to consider different approaches to capturing organisational factors in LMIC healthcare facilities. Papers were excluded if they did not clearly articulate the organisational issue, impact on provider behaviour or patient care, if patient care focused solely on clinical aspects, and if the organisational issues captured remained at the level of health systems and policy without facility-level context/detail.

### Types of sources

To identify potentially relevant documents, the following bibliographic databases were searched from 2000 to 2021: Scopus, PubMed, Google Scholar and ScienceDirect. The search

strategy was drafted by members of the review team, and further refined through team discussion. The final search strategy for PubMed and Scopus can be found in S1 Text. The final search results were exported into Zotero, and duplicates were removed by a member of the review team.

### Selection of sources of evidence

Pairs of reviewers screened titles and abstracts to identify studies that satisfied the inclusion criteria. Subsequently, a full-text evaluation was conducted by the reviewers to finalise relevant articles. A snowball search of these articles resulted in additional articles to the final list. Disagreements on study selection and data extraction were resolved by consensus and discussion with other reviewers when needed [Fig 1].

### Analysis

A data-charting form was developed by the reviewers to extract relevant variables. This was an iterative process, which involved discussing the results and updating the form. We abstracted data on article characteristics (e.g. methodology, sample), study setting (e.g., country of origin, type of facility, urban/rural setting), organisational factors captured, the impact on providers and care, and the role of resource constraints. We grouped empirical studies that focused on maternal health care and summarised them thematically. Intervention study details were extracted along similar categories and then sorted to reflect on the themes that emerged from the empirical studies focused on maternal health.

### Protocol and registration

Our protocol was drafted using the Arksey and O'Malley framework [43] along with enhancements proposed by Daudt et. al. [44] and Levac et al. [45] for different stages of the review process. The final protocol was registered with the Open Science Framework (https://osf.io/s8bpg). We followed the PRISMA extension for scoping reviews to report findings [S1 Table].

## Findings

A total of 54 articles were ultimately incorporated for analysis, of which 13 were intervention studies across all disciplines [S2 and S3 Tables]. Over 60% of the maternal health articles were from the Sub-Saharan African region [Fig 2]. The majority of the studies used qualitative methods of inquiry [Fig 3], were conducted in public facilities, and elicited provider perspectives. Over 40% of the studies focused on midwives or a combination of midwives with other cadres of providers including nursing staff, obstetricians, doctors, and facility-level managers. For the intervention studies, most were from Asia and conducted in public facilities and across multiple levels of facilities.

The organisational factors captured in the maternal health studies relate to workload, division of work, professional autonomy, pay, training, feedback and supervision, and workplace violence, with wide ranging impact on providers and care. Intervention studies drawn from other branches of medicine respond to a number of organisational challenges captured in the maternal health articles and focus on leadership, supportive supervision, peer support, responses to workplace violence, and planning for shortages.

### Part A: Organisational issues and the role of shortages

The themes that emerged from the review centre around human resource management and working conditions. While a number of these topics are influenced by upstream health systems and policy challenges, we focus on how these issues manifest at facility levels.

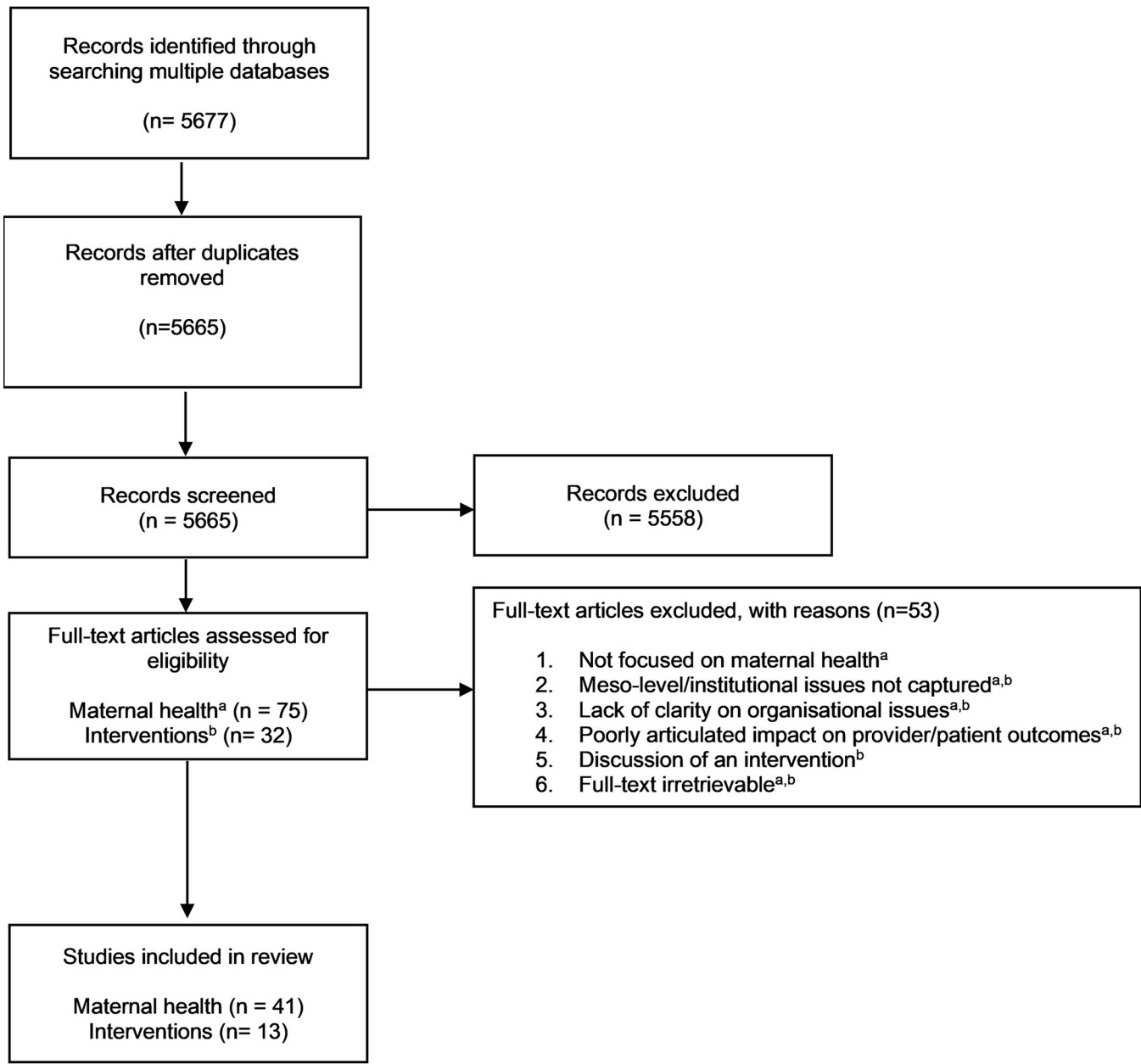

**Fig 1. Flow diagram for the scoping review process.** If accepted, production will need Fig 1 to link the reader to this figure.

We found an overarching view from providers' perceptions of management across studies, that their welfare was not a concern and that they were not taken care of [30,46–49]. This permeated several domains including workload, division of work, pay, training, supervision, and protection from violence in the workplace, and extended to broader issues of professional autonomy. In terms of the role of resource shortages on these organisational issues and care, some domains were more strongly influenced by shortages than others. Shortages in infrastructure, equipment and material supplies also impacted care directly.

**i. Excessive workloads.** Heavy workloads were the most commonly cited problem across the studies reviewed (~25/41 studies). Attending to high numbers of patients were consistently

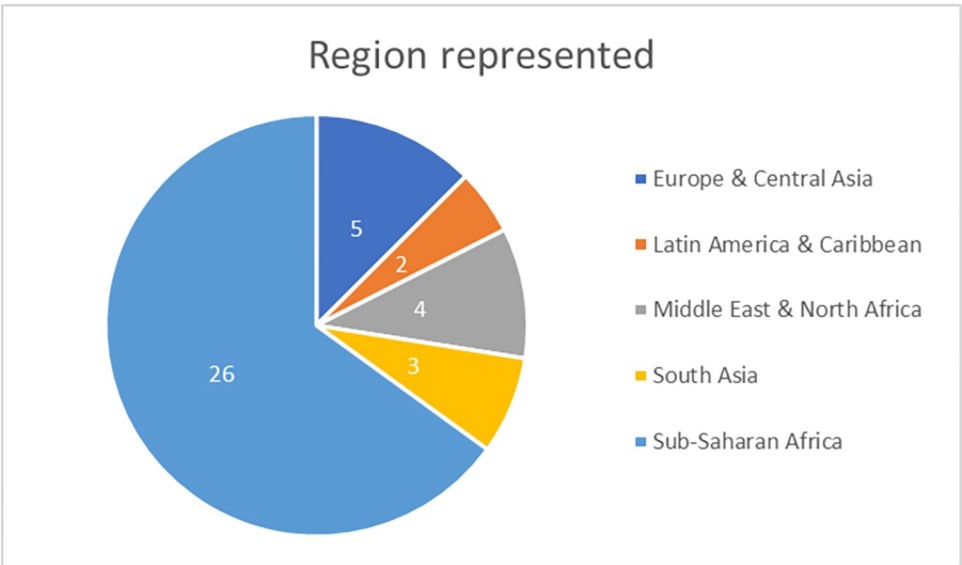

**Fig 2. Regional distribution of maternal health articles.** If accepted, production will need Fig 2 to link the reader to the figures.

reported challenges, along with back-to-back and long shifts, frequent night shifts and weekend duties. Workload pressures had the most explicit links to negative psychological effects on providers, and negative provider behaviour in LMIC maternity care settings. High patient loads also created institutional pressures for providers to ration time and attention to patients, necessarily compromising care. The pressure to "finish births quickly" also skewed care towards efficiency and medicalisation and away from women-centred approaches.

The psychological impact of excessive workloads on providers were captured widely. Feelings of stress, fatigue and exhaustion, frustration, feeling overwhelmed, and burnt out were reported across studies [50–58]. The lack of rest with heavy workloads was noted to lower staff

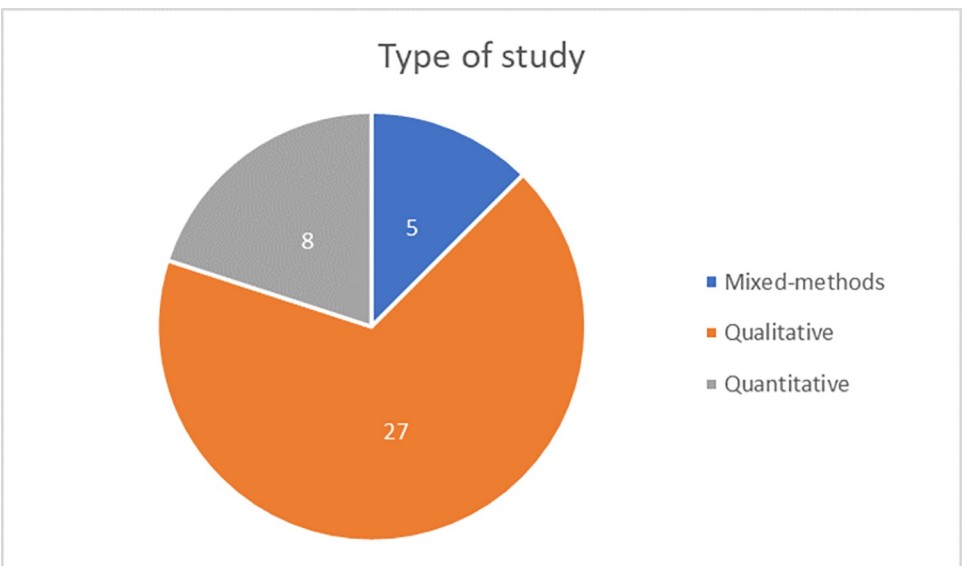

**Fig 3. Distribution of maternal health articles based on type of study.** If accepted, production will need Fig 3 to link the reader to the figures.

motivation and morale, productivity and concentration, which was linked with poorer or sub-optimal care [52,57–61].

Being overworked was also related with behaviours like reporting late to work, leaving early or making excuses to get away from the workload and sleeping in the ward when on duty [56,61–63]. Other behaviours captured relate to emotional distancing as a coping mechanism. In Kenya, nurses depersonalised care provision to cope with stress [64]. In India, midwives described a sense of detachment towards women, as Myra et. al argue, because the workload does not allow emotions to be processed [54].

Workloads were also linked with several attitudinal and behavioural problems that impinge on or directly indicate mistreatment. Providers were described as seeing communication as "a waste of time" [60], intentionally delaying care, being careless, appearing cavalier, uncaring, unsupportive, apathetic and not taking patients seriously [56,57,65,66], having tense interactions [60,66], displaying irritability and anger [54,56], and were even perceived to be "working with anger and hatred" [67]. Associations between heavy workloads and attitudes towards verbal and physical abuse have also been established. One study in Namibia found that providers dissatisfied with their workload were twice as likely to agree that sometimes pinching or slapping a woman can succeed in getting her to push harder [68]. Excessive workloads and stressful working conditions were also a major reason cited in the study to explain short-tempered and "harsh" behaviours toward women in labour.

In terms of the impact on care, across contexts heavy workloads contributed to delayed admission [69], a lack of thorough history taking and examinations [56,60,69], prolonged wait times and delays even in attention to emergencies, "taking shortcuts and skipping steps" [61] and leaving patients unattended and untreated [56] including during labour [68]. In the Nigerian study [56], patients perceived off-hand triaging to cope. This was reflected in providers attending to more serious conditions, while deferring attention to some patients, which sometimes amounted to severe neglect as described by one woman: *"it causes a problem because if a woman gives birth in the night they won't sew her [episiotomy] until the next morning; it is painful"* [56].

The pressure to "finish births quickly" and medicalisation of care as a response to heavy workloads was also noted in middle-income country contexts. An urgency to get the baby out characterised by early amniotomy, routine labour augmentation and episiotomy [52,60,69], and excessive cervical examinations to track labour progress [63] were reported in these settings. In Turkey, medicalisation as a response to workload pressures was in turn found to disempower midwives in multi-cadre teams, and distance them from women [52,60]. Even in comparatively better resourced conditions in Saudi Arabia, Abdulghani et. al. noted that staff workload pressures lead to more rushed care and less time spent on recommended practices like educating women on skin to skin with the baby, choosing medical options may be quicker instead (such as using baby warmers) [55].

*"The work is tiring, we don't have time to even talk to the mother. I hardly take a break for prayer or lunch. I'm not saying that we could not perform the practice of skin-to-skin contact, it is easy and simple but with all other competing tasks we find it is hard to do. You cannot understand the situation until you work here and experience being under pressure. The problem is that we reach a stage where we just work blindly like a machine. We would love to spend time with the mother and educate them but being busy all the time makes us tired and counting the time to go home."* Obstetrician, [55].

Resource shortages, particularly inadequate staffing, directly impacted workload. In fact, most studies and provider views of workloads described them synonymously with staff

shortages, with similar outcomes, including lapses in care provision, medicalisation, and negative psychological effects on providers. In Malawi, Bradley et. al. noted that clinical officers, who were in short supply, struggled to prioritise whom to treat, resulting in interruptions in care and tasks being left incomplete [61]. Some contexts also reported providers being responsible for managing both labour and postnatal wards at once [49,61]. In Turkey, *"....midwives pointed out that non-medicalised birth is a time-consuming process that lasts, on average, up to 24 hours. Due to a lack of available health care personnel, there is not enough time to follow up with the women"* [60] Confronting the negative impact of shortages on care was also found to cause feelings of guilt, sadness, and demoralisation among providers [70]. Hiring temporary midwives in a South African teaching hospital was not found to sufficiently address the work burden because they still needed to be oriented and supervised [53].

Teasing out the role of staff shortages in the maternal health studies beyond increasing workloads, however, points to fundamental problems in how work is divided and worsened by staff shortages.

**ii. Unequal and inefficient distribution of labour.**   Across a wide range of settings, unequal and irrational distribution of work between cadres was a reported organisational challenge [30,51,56,57,62,63,70–73]. In Kenya, Warren et. al noted that poorly managed duty rosters did not maximise staffing ratios and contributed to high workloads [71]. Similar issues related to poor allocation and management of duty rosters and rotation schedules were also highlighted characteristics in a Mexican teaching hospital [63]. Nurses in Malawi complained of rotation practices that expected that they would be skilled enough to serve the distinct requirements of different wards [61]. These weaknesses resulted in both overburdened and under-utilised 'lower-level' staff—especially midwives and nurses.

Teaching hospitals in particular were shown to underutilise nursing and midwifery skills, revealing the influence of medical and institutional hierarchies in shaping how work is divided. Medical staff in a large teaching hospital in India did not collaborate with nurses, who were largely kept out of medical work and relegated to administrative tasks, despite workload pressures [30]. In an Ethiopian teaching hospital, frequent turnover of interns and general practitioners made midwives the more stable cadre with significant expertise, yet they were responsible mainly for sterilisation of equipment and maintenance of the delivery room. Meanwhile it was reported that too many interns crowded around women delivering [70]. In the DRC, midwives reported being placed at departments where their midwifery skills were not utilised, such as surgery/internal medicine wards, the pharmacy, or the administration office [74]. They also noted being shifted across departments according to their supervisor's preferences rather than their professional competence. While Mozambican midwives in Pettersson et. al.'s study appealed for rational distribution of work with rules to ensure providers help each other and assist in areas where it is most needed, they emphasised the importance of buy-in from senior staff to ensure it happens [73]. Irrational work distribution was seen among junior doctors as well. In a Mexican teaching hospital, first year residents were assigned to "grunt work" and full-time physicians managed the ward and complications. In this setting, senior physicians were observed sitting by, chatting and joking in the ward, while junior residents worked without respite [63].

Staff shortages were shown to exacerbate the poor utilisation of nurses and midwives, diverting available skills away from direct care. In the Malawian context cited [61], nurses had to spend time securing care from doctors for obstetric complications. In a Palestinian referral hospital, midwives spent time in search of supplies and assisting doctors, instead of caring for women [60]. In the context of a shortage of midwives in South African facilities, midwives were reluctant to use available interns to monitor women who wish to move around in labour, because interns were expected to focus on building skills related to delivery [75].

Shortages of more skilled personnel led to ad-hoc task shifting, forcing providers to perform beyond their training [68], which directly risks patient safety. For instance, a shortage of clinical officers and doctors in Malawi meant that the responsibility of handling complications was shifted to nurses [61]. Shortages among lower cadres, pushed tasks on to support staff, patients or their families. In a study from Malawi, a shortage of midwives led to cleaners being asked to attend to labouring women during their breaks [57]. Shortages of support staff and cleaners have contributed to patient families' having to carry out tasks [30,62] like wheeling patients and delivering samples and medicines [30], or patients themselves being expected to clean themselves or the newborn after giving birth [56,68]. In an Indian hospital, the need for assistance from families for such tasks led to overcrowding of male relatives which invaded women's privacy during labour [30]. Ultimately, shortages of staff appeared to worsen disparities in how work is divided, disproportionately impacting those cadres "lower" in the medical hierarchy while further compromising care.

**iii. Lack of professional autonomy.** Closely linked with and underpinning issues of poor work distribution were challenges related to professional hierarchies. Expectedly, most issues related to inequalities in power, autonomy and decision-making appeared to be concentrated around junior doctors in teaching hospitals [30,63,76], and nurses and midwives across settings [52,53,66,73].

A number of studies highlighted how reduced autonomy over practice impacts provider wellbeing and ultimately care provision, particularly for nurses and midwives.

Turkish midwives expressed that though they see their role as managing births, they are forced to take a back seat to doctors [52]. They also felt demoralised when they were supervised by clinical officers who had lesser experience and had the ultimate decision-making powers in cases. *"Sometimes you call for a clinician. . .they have not been with the mother. . .and your opinions [on management] collide. The midwife has spent more time in school, but the clinician has the final say. . .there is nothing you can do. It is frustrating as you only want the mother to benefit."* [52]. Conflicts with physicians were also among key workplace stressors reported by Slovakian midwives, sometimes resulting in 'mal-adaptive coping mechanisms' such as behavioural disengagement and venting [77].

For midwives across settings, not being allowed to practise their profession, not receiving recognition for their work, being unable to influence decisions made by more senior staff or management and experiencing professional disrespect and inferiority within multi-cadre teams, in combination with resource constraints, led to stress, frustration, feeling undervalued, demotivated, demoralised to the point of making it challenging to continue their job [73,74,78]. Being unable to independently take actions that are in the best interest of patients was also argued to contribute to moral distress among nurses in Egypt [79] and Malawi [62]. *"We tried to discuss problems several times, but we have been always told: you have no right to discuss anything. Do not discuss. Your duty is only to receive and admit patients and work with them. If you don't like it here, just leave work and go home. Or we got no response."* Nurse, [60].

Some studies pointed to how professional marginalisation can translate to irrational care practices and mistreatment. Madhiwala et. al. described hospital structures and processes that operate based on hierarchies and authority over standards and protocols. Junior providers in this context were left out of decision-making, and irrational, harmful practices endorsed by senior staff were reproduced and institutionalised [30]. In South Africa, midwives reported that their expertise was side-lined by doctors, and believed that interventions like caesarean-sections could be reduced if they had more say [53]. Palestinian midwives reported that they were shouted at for following practices that are aligned with midwifery care but go against ward policy, such as not routinely starting an intravenous line for labouring women or allowing women to walk during labour [60]. Peterson et. al. make the link to the mistreatment,

highlighting that *"midwives who claimed to experience demeaning and frightening intra- and interprofessional communication justified the use of fear-arousing communication as a tool to secure women's collaboration"* [73]. Similarly, Myra et. al. also argue that *"being voiceless in planning care provision leads to an assertion of power over women who are further down in the social hierarchy"* [54].

In the face of staff shortages, not empowering midwives and nurses to practice the full scope of their clinical training led to deviations in practice norms. *"The midwives gave examples where the doctors failed to attend cases during the night shifts, and delayed the suturing of episiotomies, leaving women waiting in the lithotomy position for a long time. Some midwives take the initiative and suture episiotomies themselves instead of keeping the woman waiting. It should be noted that the Ministry of Health has decreed that midwives are not allowed to suture episiotomies or perineal tears, despite the fact that they are trained in this procedure"* reflecting inconsistencies between policy and professional education that impact care provision [46]. Madhiwala et. al. also noted nurses stepping in to handle deliveries out of necessity when doctors were unavailable though they were not officially empowered to do so in referral hospitals in India [30].

**iv. Inadequacies in training.** Facilitating continuing medical education (CME) and in-service training was included in our review as it falls within the ambit of organisational management. We found poor access to training opportunities, and poor training impacting provider confidence, competence, and care practices [32,47,49,50,53–58].

As training is linked to career advancement, a lack of training opportunities for practicing midwives was found to contribute to poor motivation and work performance [49,51,73]. In Tanzania, midwives noted differential access to the few training opportunities available, with "those who work in administration" prioritised [49]. Midwives in Mozambique felt "a lack of competency, inexperience, and ability" which they linked with their low status within an obstetric team, reporting frustrations around professional inadequacy and advocating for additional practical and theoretical training [73].

In terms of the impact on care, in rural [59] and urban [51] Ghana, lack of training opportunities was not surprisingly, found to influence adherence to evidence-based practices and quality of care. In a Jordanian hospital, lack of CME for breastfeeding was linked with poor knowledge, attitudes and support for breastfeeding, and ad-hoc advice [66]. Inadequate clinical skills were linked with inappropriate care, including mismanagement of pain among Namibian maternity care providers [68]. In a Dominican teaching hospital, inexperienced and inadequately trained residents oversaw care provided by medical students, interns and nurses. While senior providers did walk through the wards, they were rarely observed to provide direct care or hands on training there. Extensive deviation from norms on clinical management, rational obstetric practice, and professional standards were documented in this setting [76].

Another dimension of on-the-job training is guidance on clinical norms and protocols. As Miller et. al. found, poor orientation of norms and guidance on how to apply them impacted quality of care in a teaching hospital setting [76]. A provider from Warren et. al.'s study also reported that poor supervision of interns resulted in malpractice and excessive vaginal examinations during labour [71].

Poor role orientation may also be seen as an extension of poor training. A Tanzanian study also noted that job descriptions of obstetric staff were found to be generic and did not reflect the actual demands of the role. In a publicly funded Turkish hospital, doctors did not see providing emotional support as part of their role [69].

"That kind of support could be given by psychologists or specially trained midwives.

*Maybe doctors could give it, but doctors have to attend births, perform caesareans, check on post-op patients, write prescriptions, and visit the wards."* (Ob/Gyn specialist) [69]

For midwives, not feeling competent combined with difficult working conditions resulted in their undertaking considerable personal responsibility and feeling inadequate at work [51]. Facility-level shortages may exacerbate the problem of poor access to training; staff shortages were noted to prevent existing staff from taking up continuing medical education opportunities such as attending workshops and trainings [49,53] that aim to address the skill gap and build capacities of frontline providers.

**v. Pay-linked dissatisfaction.** Problems resulting from poor pay were cited across studies and especially among midwives [49–53]. In the Democratic Republic of Congo (DRC), not having an equitable remuneration system, reflected in low pay and irregular payment, made midwives feel underappreciated. It also prompted midwives to take up second jobs which increased their work stress [74]. Not being compensated for working overtime was also reported by midwives in Tanzania, which contributed to feeling devalued in their work [49]. The study also highlighted the importance placed on smaller scale financial support such as travel allowances and food, which were not offered. In a study that highlighted the position of midwives in Ghana, participants indicated that although other health providers, such as doctors, were given rural allowances as incentives to accept posting to rural and deprived areas of northern Ghana, midwives were not [59]. Favouritism and a lack of transparency [32,47,62] in access to training and for salary top up allowances, disbursements of salary, advances and loans also characterised relationships between providers, their superiors and management across settings. This generated distrust among nurses, midwives and management and contributed to demotivation and intentions to leave [47].

A few studies reported the influence of pay on the way providers treat patients. In a survey of providers across 35 district and referral hospitals in Namibia, the vast majority reported being dissatisfied with their pay, particularly when compared to their workload [68]. Qualitative findings from the study supported that this led to being unfriendly towards patients, with one provider stating *"If we [were] paid according to the patients that we care for, you will see us being friendly"* [68]. In another direct link to mistreatment, low salary was cited as the reason why Afghan providers asked for bribes in public clinics where services are meant to be free, claiming that they need the extra income to be able to support their families [50].

In terms of the influence of shortages, most studies noted poor pay along with other difficult working conditions, like heavy workloads and lack of infrastructure, contributing to dissatisfaction and poor morale. While poor pay is ultimately determined by upstream funding constraints that were beyond the scope of this review, we can establish that problems with pay only worsen in poor facility conditions, as providers find that gap between their pay and actual workload even harder to reconcile.

**vi. Opaque feedback processes and poor supervision.** From the large proportion of provider perspectives represented in the studies reviewed, it was challenging to distinguish between problems providers faced with facility-level managers who may periodically evaluate overall performance, and clinical supervisors who are meant to routinely oversee how care is provided. Nonetheless, a number of studies identified challenges across these areas [30,32,47,59,60,62,65,70,71,74–77]. The absence of formal appraisal, feedback and communication about job performance emerged as a key feature of poor management. In a study of moral distress among obstetric nurses in Egypt, Hassan et. al. found an absence of formal meetings to guide, audit or provide feedback on work practices [60]. In a Palestinian referral hospital, evaluations of providers were identified as lacking openness and objective criteria, not being participatory or based on merit. *"Providers did not know the basis for the evaluation, informed of their strengths and weaknesses, or given the chance to improve"* [60]. In Malawi and

Tanzania, a lack of positive acknowledgement for job performance for several months [47] and even years [32] contributed to low motivation among providers. Further, perceiving facility managers to be unsupportive, providers did not value their feedback [32]. A lack of responsiveness from management to requests or suggestions that would impact patient outcomes led to demoralisation among obstetric staff in Malawi [47].

In terms of the implications for care, an Armenian study of nurse-midwives identified that performance reviews and feedback almost never related to staff behaviour [80], which would be a component to prevent mistreatment. Chipeta et. al.'s study [47] echoed other studies in the need for open communication between staff and managers, highlighting the detrimental effects on care provision when appraisal processes focus on negative feedback. *". . .it really affected my performance. I would say for about 2 days I didn't touch a patient. . .If you are demotivated, you don't have a feeling to work and at the end you find out that the patients are the sufferers."* [47]

Other problems related specifically to poor clinical supervision, which was linked in studies to non-adherence to standards and being unavailable to patients. Nurses in new-born units in Kenya were less likely to remind mothers to visit their babies to feed them at night, resulting in night feeds being more likely to be skipped. McKnight et. al. attribute this to lower expectations of providers at night [64]. In another Kenyan study, poor supervision meant providers took long breaks or skipped night shifts altogether [71]. As noted earlier, poor supervision and guidance of interns in particular in different settings [71,76] was found to cause deviations in professional standards that amount to mistreatment, such as excessive vaginal examinations.

The absence of supervision was not the only challenge. Across numerous studies, when supervision was provided, it centred around criticism, fault-finding, blame, ridicule, and punitive responses to errors [30,32,47,53,55,60,61,81]. Chipeta et. al. highlighted several negative characteristics to the supervision providers received, as one clinical officer explained *"People have left the hospital, people have joined NGOs, because of the attitudes towards new recruits. . .the way they speak and the way they supervise you is more of a picking somebody. . .or picking on your personal weaknesses. . .they want to show their superiority by intimidating others."* [31].

Midwives and nurses in two large public hospitals in Saudi Arabia reported negative comments and pressure from senior providers (OBGYNs) impacting adherence to evidence-based practice and patient trust. *"I practice mother-infant skin-to-skin contact and I know it is evidence-based practice, but I practice when I am alone with the mother because I am afraid, I will get negative comments like "finish your job quickly" or "it is your responsibility if the baby falls down". I feel reluctant to do the practice. The problem if they said that in front the mother, this mother will not trust me or my work."* [55].

In Palestinian hospitals, punishments dispensed to nurses and midwives for making errors included having off-duty hours reduced or denying a day off [60], only worsening the conditions that contribute to errors. In Malawi, threats from managers over what were perceived as small errors by obstetric staff contributed to a lack of interest in work, withdrawal and unwillingness to report to work [47]. In a survey of 57 providers in Ethiopia, 40% felt they were poor or very poorly supported by their facility management [48]. These views on management were documented along with observations of various forms of mistreatment of women reported by Asefa et. al. [48] and others [46], though their associations were not explored in these studies.

Madhiwala et. al. illustrate how a "supervision system based on attributing blame" percolates across inter-professional relations and down the medical hierarchy causing stress and anxiety among lower level providers and a creating a harmful organsiational culture. In a referral hospital setting in India, this translated to residents feeling anxiety about being wrong and they in turn engaged in correcting rather than mentoring junior staff. Medical staff did not

collaborate with nurses, and nurses shouted at and reprimanded support staff [30]. Bradley et al. note that a culture of blame also impacted temporary staff from carrying out procedures they are trained to do, reporting that they lacked the confidence and feared being blamed if something went wrong [61]. A lack of supportive supervision therefore may also impair the ability of temporary staff to lower and rationalise workloads in facilities.

We found some evidence of how shortages of various kinds may influence supervision. Infrequent supervision visits from higher level management impacted regular supply and maintenance of drugs and equipment across facilities in rural Tanzania, but financial constraints were cited by managers as a reason for low contact with facilities [82]. For providers in Malawi, having requests for supplies ignored was also linked with low morale, though systematic shortages may underlie this [47]. Providers also felt poorly supported by supervisors when dealing with lapses in care that may be precipitated by shortages. Providers from another Malawian study expressed *". . .our bosses do not back us when there is a problem, whilst they are very aware that the cause of the problem is really the staff shortage"* [61].

**vii. Lack of institutional protection from workplace violence.** Multiple studies noted workplace violence, characterised by anger and aggression from patient families, sometimes patients themselves, and from other health workers impacting providers [30,48,49,59,61]. A study in Ethiopia that documented widespread mistreatment of women found that well over half of providers surveyed reported that they themselves had been disrespected and abused by other healthcare providers or clients [48]. In India, providers, especially subordinate staff, noted that they were unsupported by management to provide care equitably, citing cases where complaints were filed against them for denying preferential treatment. It was emphasised that they did not have institutional safeguards to support and protect them when they felt threatened by some patient's families, prompting them to seek police protection instead [30]. Similarly, Samir et. al noted a lack of formal processes to report violence from both colleagues and patient's families [65]. Providers not having institutional mechanisms to prevent, report and respond to workplace violence contributed to an overall sense of being unsupported by management and that their welfare was not a concern. Other evidence highlighted the implications of workplace violence on care provision. Midwives in Tanzania reported being abused by their patients, which they felt undermined their efforts to provide care. They found it lowered job satisfaction, led to increased errors, and decreased quality of care [49].

As for the role of shortages, Samir et. al. observed that resource shortages may underlie psychological and physical violence against nurses in Egypt [65]. Hassan-Bittar et. al. also highlighted how nurses and midwives may be more vulnerable to anger from patient's families as they are faced with inadequacies in care such as beds or doctors being unavailable [60]. While these conflicts may be linked with how shortages compromise care, it also highlights how frontline providers, who are typically lower in the medical hierarchy may be at greater risk of workplace violence.

**viii. Infrastructural inadequacies.** *Other challenges in working conditions*. It is important to note that shortages also directly impacted care, with a degree of independence from other issues in organisational management discussed. Resource shortages were noted in a majority of studies reviewed (30/41), and not limited to public or publicly-funded facilities alone. These took the form of infrastructural constraints such as limited space, small birthing units, and a lack of beds [51,57–59,69,70,73,74,82,83]; inadequate electricity and water supply [67,71,74,75,83,84]; and a shortage of equipment, drugs and supplies [46,49–51,57,59,64,67,70,71,73–76,80,82–84].

The lack of beds in particular resulted in conditions where women laboured and/or delivered on the floor, in corridors or had to share beds [30,54,57,69]. I Additionally, congested wards and not having curtains or screens in place were noted across regions, violating

women's privacy during birth [30,54,57,69]. Limited space was also the reason cited by providers across studies to deny birth companions [30,57,70]. *". . .I guess we were quick to preach about companionship at the time of birth. We need to work on our infrastructure before we start advocating for companionship during birth."* [57].

Overcrowded labour wards created institutional pressures to clear beds, which were seen to support practices such as early amniotomy, excessive cervical examinations, routine augmentation and episiotomy, and high rates of c-sections [52,63,69]. Other limitations of poorly organised spaces were highlighted in the context of staff shortages and high patient loads [30,73]. A midwife in Pettersson et. al.'s study noted the distance between the labour ward and other workstations. As the only provider in the labour ward, calling for help required leaving the labour ward, putting delivering women or babies at risk. In the context of staff shortages, *". . .restructuring of the labor ward was felt to be the most appropriate action to ensure that the midwife had simultaneous access to the woman and her newborn infant."* [73]. Madhiwala et. al. noted that the design of the labour ward in a teaching hospital did not allow for the presence of a labour companion [30].

Apart from the shortage of space and beds, shortages of medicines, equipment, and supplies was clearly found to constrain providers, making it difficult for them to do their job or work effectively, and ensure quality [46,50,71,75,84]. These conditions ranged from not having crucial medicines including anaesthetics (resulting in women having to endure episiotomy and suturing without pain relief), coping with poorly maintained equipment such as blunt episiotomy scissors, and lacking basic protective gear, among other shortages. *"Always we do not have urinary catheters in stock so we compromise by using suctioning catheters for emptying of the bladder. We are aware that the suctioning catheters are hard not soft in comparison to the urinary catheter and that it is easy to injure the patient. But what can we do?"* [75].

In an Ethiopian referral hospital, providers had to scramble to assemble equipment, find basic necessities such as gloves, and could not efficiently use their time during preparation for delivery [70]. Multiple studies [49,59,82] noted that the lack of supplies involved greater infection risk, endangering both providers and women. They also highlighted that such constraints required providers to adapt and improvise to provide care, which ultimately led to sub-standard care [59,82]. In terms of impeding recommended practices, Shattanawi noted a lack appropriate infrastructure like refrigerators for pumped milk among factors that inhibited breastfeeding and its promotion among preterm infants in an NICU in Jordan [66].

In a study of Ghanaian midwifery schools, nearly 70% of midwifery students said the resources they have available to them influence how they treat patients [46]. The burden of working without adequate supplies was also found to frustrate providers, make them feel insecure and not taken care of in their workplace [49], and resulted in low job satisfaction [84]. The pressure these constraints put on the patient-provider relationship may be two-fold. In rural Nigeria, Asuquo et. al. noted that women were shouted at for not bringing supplies with them [84]. Two Tanzanian studies reported that providers were blamed by patients' families for the lack of supplies, and accused of bribery when they asked for supplies [49,82], which can erode trust between providers and patients, and communities.

*"Since policy states that delivery services are free, the midwives reported that asking for equipment was often interpreted by the clients as asking for money for the midwives' own personal use."* [49] We also know that compromised care due to such shortages may trigger aggression towards providers that constitutes workplace violence [60,65].

Among the non-intervention studies focussed on maternal healthcare, issues relating to workload and supervision were captured to a greater degree compared to problems of pay and training. This may be because pay and training are defined by larger health system and policy priorities, and we selected papers that described organisational and working conditions at the

facility level. We nonetheless identified a large number of organisational issues that impact providers and impinge on care in myriad ways. Workload was most strongly influenced by staff shortages, with some influence seen over access to training, pay (though moderated by workload), and division of work. Shortages in other areas of infrastructure and supplies were shown to impact care directly. Autonomy over practice and supervision were least sensitive to shortages based on the papers reviewed. Importantly, we found medical hierarchies in institutions operate as a meta theme, impacting almost all areas of organisational management. We see its influence most clearly in terms of how work is divided, approaches to supervision and restrictions on professional autonomy. But we also see it underlie deficiencies in training and renumeration. In sum, we find that although resource shortages are a major challenge in LMIC maternity care settings, there are other important areas of organisational management at the facility level that impinge on provider behaviour and care.

The following section describes interventions of interest that speak to one or more organisational challenges discussed.

## Part B: Responses to organisational challenges

Based on the organisational themes that emerged from the analysis of maternal health studies, the included intervention studies below capture changes in organisational management that are of relevance to RMC.

**1. Planning for pressures and shortages.** Responses to human resource shortages in the public sector have included outsourcing. Our results yielded one study [72] that assessed the impact of outsourcing nursing staff in two departments in an Iranian hospital. The pre and post study found that while the staff per bed ratio increased, there wasn't a notable increase in satisfaction with outsourced staff among administrators, managers and supervisors. The study was limited in providing detail on why satisfaction with outsourced staff was low, how it impacted existing staff or workload. It is worth noting that the study was published in 2001—the health system context and experience with outsourcing services in Iran may have evolved considerably since.

One of the outcomes of disproportionately heavy workloads, is the stress on providers to manage obstetric emergencies along with routine care [61]. Creating 'maternity waiting homes' near rural facilities has been a widely adopted approach to improve responsiveness to obstetric emergencies in low resource settings. Kaiser et. al.'s assessment of how maternity waiting homes in rural Zambia affect the health workforce and maternal health service delivery demonstrates that changes to the structure and process of services can result in improvements in organisational management, even in the event of increased workloads [85]. With maternity waiting homes enabling women to arrive early and stay longer post birth, providers did report added responsibilities in the context of existing staff shortages. However, the presence of these homes also helped staff improve the planning of their work, which helped them provide better, more timely care. This, in turn, contributed to providers feeling "better within their roles".

The review of maternal health studies highlighted the impact of shortages of medicines, supplies and equipment on providers and care. Trap et. al [86] describe an approach to reduce those pressures through better supervision. The intervention trained pharmacy technicians and pharmacists in stock management and supervising clinical providers in adherence to standard treatment guidelines (STGs). It found that training and supervision of these cadres can improve management of medicines and drugs and increase the rational use of drugs in the context of shortages.

**2. Reshaping leadership.** The following intervention studies respond to several issues linked to hierarchies found in the maternal health studies, including poor communication

between providers and with management, a lack of transparency and favouritism, and a culture of blame.

Tuan's study of a clinical governance intervention in a public hospital in Vietnam documented how a change in leadership marked by a new value system, based on improved knowledge-sharing and trust, among other factors, reshaped professional (inter-cadre) and provider-patient relationships. The methods involved triangulating data from hospital documents, observations, and in-depth interviews. The study, however, lacked detail on the actual components of the intervention or objectively measured outcomes. It also did not address how the new approach helped providers cope with existing constraints. Nonetheless, it argues that leadership focused on improving knowledge-sharing and trust at the facility-level can foster support and understanding between providers and create greater patience and empathy in care provision [87].

Hee Jeon et. al's evaluation concerned an ethical leadership program for South Korean nurse managers. The six-month training intervention involved a combination of (a) lectures and reflective group discussions, (b) practice and planning of ethical leadership activities (including peer mentoring), and applied tasks. Their pre and post study results demonstrated statistically significant improvements in the domain of 'people orientation'. They also found that the domain relating to sharing power improved significantly only for early career managers (less than 5 years of experience). The authors suggest that this may be linked to early career managers being more motivated and proactive about learning new skills. The study also found that staff nurses' perceptions of their unit manager's ethical leadership were significantly related to job satisfaction, organisational citizenship behaviour (discretionary actions that go beyond one's job role), trust among peers and with supervisors, and a domain related to justice and respect [88].

An action research project in South Africa aimed to change organisational culture in a community health centre by focussing on transforming leadership. A cooperative inquiry group (CIG) was formed comprising administrative, clinical and management staff within the facility. They focussed on identifying issues related to organisational culture, which at baseline was characterised by hierarchy, control, blame, poor recognition, low transparency and information sharing, cost/resource reduction, long hours, among other factors. Over an 18-month period, the CIG held meetings to plan, implement and reflect on organisational changes. Three key leaders in the facility also went through a leadership values assessment and six months of coaching based on feedback. Cultural entropy ("the level of dysfunction in an organisation resulting from limiting beliefs and fear-based behaviors of leaders") reduced considerably based on pre and post test measures. The culture moved towards greater communication, appreciation, accountability, teamwork, and patient orientation among other positive changes. Importantly, the study showed that organisational transformation is possible even in the context of resource constraints and heavy workloads [89].

**3. Providing supportive supervision.** Several issues around supervision were identified in our review of maternal health studies, chiefly that supervision was either inadequate or absent, or when provided, it focused on punitive responses, fault-finding, and blame. The interventions below promote structured and supportive feedback and supervision.

In 'Patient safety walkarounds' (PSWRs), hospital managers engage in regular, structured walks through the hospital to allow frontline staff to report safety concerns and their causes.

Saadati et. al. [90] assessed a PSWR intervention in an Iranian hospital by triangulating all documentation related to PSWRs over a five-year period and carrying out a content analysis. The approach reflected a way to capture safety concerns emanating from the work environment that traditional error reporting does not enable. In addition to improving the

identification of patient safety incidents and their resolution, the approach was shown to improve teamwork and open communication between providers and management [90].

Uduma et. al. [91] used a randomised experimental design to evaluate the impact of a supportive supervision programme in Tanzania. It comprised a series of workshops on human resource management, a five-day intensive training on supervisory and support skills, and an additional component of action learning that continued for 12 months. Based on supervisors' self-assessment at endline, a majority of supervisors felt that they were much better at 'treating staff with respect and recognising their contribution', and an improvement in 'problem solving within the facility'. The least improvement was seen regarding the overall workload. Health workers also assessed changes and indicated better supervision processes, although these improvements were less marked than supervisors' self-assessment. Other changes included shorter but more frequent supervision by both groups. The power of the study was limited by the fact that over half the sample changed between baseline and endline but the intervention nonetheless helped "remove some self-perceived barriers such as time management and lack of confidence" and improved understanding and application of supportive supervision practices [91].

'Patient safety culture' comprises several important dimensions of organisational culture that are relevant to enabling RMC. These include open communication, non-punitive response to error, positive reinforcement for following standards, teamwork (mutual support and respect), management being receptive to staff suggestions, among other areas. Xie et. al. studied the impact of a safety culture training program for nurse managers in five public hospitals in China. Post training, there were statistically significant improvements in managers' perceptions of patient safety culture across several dimensions mentioned above [92].

**4. Boosting resilience through peer support.**   Poor feedback and communication, combined with difficult working conditions, contributed to low morale and other areas of performance in the maternal health studies reviewed. The following intervention involves an approach to boosting resilience and morale through peer support.

A program designed to empower and enhance resilience and organisational commitment among new nurses in Korea was assessed thorough a randomised controlled trial in two hospitals. The intervention involved a combination of an off-site reflective session, a post work 'huddling' programme where nurses discussed negative feelings associated with job stress, workload and interpersonal relationships (moderated and guided by a mentor), and a smartphone based social networking service that sent messages related to mutual encouragement and positive reinforcement. Standardised questionnaires were used to assess the effect of the intervention. Though measures of commitment and empowerment were higher in the experimental group, and post-test turnover rates were lower in this group, the intervention did not impact measures of resilience significantly. It nonetheless highlights the value of peer support programmes for the management of new nurses, and the value of having safe spaces to discuss work stressors [93].

**5. Mitigating workplace violence.**   Workplace violence, which includes interprofessional conflicts and those with patients and/or their families emerged as a challenge in working environments in the studies reviewed. Evidence suggests that conflict with patient families in maternity settings are compounded by resource shortages and that providers feel poorly supported by management when conflicts arise. Three intervention studies related to workplace violence cover both individual responses and process changes to respond to the problem.

Approaches to reduce workplace violence were focused on nurses, across multiple departments including maternity care. Education and/or training programs, risk assessment checklists and prevention protocols were some of the methods used to reduce workplace violence [94–96]. Nurses were trained in early identification and grading the risk of violence from

patients and their families [94]. Capacity building resulted in positive attitudinal change among nurses to cope and respond to patient/family aggression [95,96]. The interventions did not focus on ways that managers can support providers and respond to escalating events. Al-Ali et. al. emphasise that organisational commitment in providing a safe environment for providers and enabling them to voice their concerns is vital in conflict management [95].

The descriptions of interventions outline responses to several problems with organisational management identified in our review of maternal health studies that impinge on RMC. Some interventions focussed on the 'hardware' aspects outlined by Gilson et. al. [42], of better planning and organising [85], resourcing [70] and management of existing resources [86]. These interventions, relating to better management of existing workloads and resource shortages were linked with improved provider morale and rational practice that concern RMC. The larger proportion of interventions engaged with the software elements of values, norms, and relationships [42]—improving relationships between providers and with management through supportive supervision, open communication, ethical leadership, boosting emotional support through peer networks, and fostering inclusive decision-making. Though only one intervention study reported on empathy and patience in care provision as an outcome [87], other studies reported on factors like job satisfaction, stress, resilience that can influence positive attitudes and behaviours towards patients.

The organisational issues and strategies identified from the studies reviewed are summarised in Fig 4.

**Fig 4. Organisational issues and corresponding strategies.** If accepted, production will need Fig 4 to link the reader to this figure.

## Discussion

Our review was concerned with how organisational challenges impact providers and care, and how they are influenced by or a response to resource shortages. We found that challenges stemming from poor organisational management interact with each other and with resource shortages in complex ways that have important implications for RMC. On the whole, shortages appeared to exacerbate shortcomings in or the effects of poor organisational management to diminish motivation and morale across cadres, distort obstetric practice, and lower quality of care. But this is not the full picture. The effects of shortages were not always powerful or even across management domains. For example, negative supervision practices expressed through punitive responses to errors and a culture of blame appeared to be minimally influenced by resource shortages, as were constraints on professional autonomy.

Issues stemming from institutional and medical hierarchies may be insidious and pervasive, independent of the presence of shortages. We found that across levels of facilities and country contexts, processes related to allocation of work, training, remuneration, and supervision, reflected and reinforced inequalities tied to professional status. These findings are supported by a 2016 global consultation on midwifery care [97]. Our review indicates that shortages of staff and other resources intensify how medical hierarchies operate, disproportionately impacting those at the bottom.

Professional hierarchies may lead to poor deployment of skills among 'lower' cadres. In larger institutions, and especially teaching hospitals, nurses and midwives may be assigned away from direct intrapartum care. Such apparently perverse work distributions in the presence of shortages of trained staff for intrapartum care point to how the effect of hierarchies can trump a rational response to resource constraints. Given staffing constraints among doctors, nurse or midwife cadres may be compelled to fill gaps in care provision at great professional risk, or choose not to overstep professional boundaries, denying women care. It is also not surprising that nurses and midwives operating in such grey zones have tense and sometimes hostile interprofessional relationships with doctors linked to their scope of practice. Not being able to do what is right for patients, due to a combination of resource shortages, lack of professional empowerment, or poor organisation and cooperation between cadres, can lead to low morale, moral distress and burnout. Such psychological responses are linked with behaviours such as disengagement and avoidance of patients that impinge on respectful care [18,21,62,79].

Power inequalities in institutions can also translate to negative supervision practices by clinical and managerial supervisors. In our studies, the practice environments of frontline providers were heavily characterised by fear and anxiety about being held individually responsible for poor outcomes, rather than reflecting a culture of shared responsibility and accountability for patient wellbeing. Resource shortages may compound this issue by prompting providers to practice and behave defensively to prevent poor birth outcomes [98]. Ramsey's recent work on mistreatment reflects a number of the organisational issues identified here, explaining that "providers were overstretched and working in risky environments and therefore made choices, conscious or unconscious, to ration the emotion work required to care for patients." [99] Other studies in low-resource settings clearly point to the positive influence that organisational management can have in this area. In their comparison of two tertiary hospitals, Tibandebage et. al. demonstrate that "management practices that increase the empowerment of nurse-midwives by making them feel supported, valued and rewarded while maintaining firm rules around bad behavior can result in higher quality care even with constraints. Empowering management practices include participatory management, supportive supervision, better incentives, and clear leadership concerning ward culture." [81]

Among the intervention studies reviewed, efforts to change organisational culture in South Africa [89] appear to have addressed the problems of poor leadership despite resource constraints and heavy workloads by fostering greater communication and teamwork across cadres. Other interventions designed to encourage greater communication between providers and management [90] to promote supportive supervision [91] and safety culture [92] also suggest that health system software can be successfully reoriented, even if the problem of skewed workloads remain unaddressed [91]. These efforts are in line with a growing discourse on 'compassionate care', which places provider welfare at the centre of solutions, emphasising the need for facility level leadership and management support to improve provider well-being and morale and promote respect [100]. This is also reflected in an early intervention study in Kenya, designed to reduce mistreatment and promote RMC, which focused on "caring for the carer" and staff wellness and improving facility-level management [26].

What we do not know, however, is how long these changes can be expected to last if they are not institutionalised; or how the dependencies and the obligations of individual facilities towards the larger health system may complicate newly built relationships of trust. We also do not know how long such reforms can survive if resource constraint problems and skewed workloads remain unaddressed for long periods.

The review also raises questions on the structures and processes of power. Hee Jeon et al's evaluation [88] suggests that the concentration of power is more difficult to dislodge among senior staff who benefit the most from the professional hierarchy. Uduma et al's evaluation [91] indicate that health workers may not completely endorse their managers' perceptions of improved supervision. And the intervention designed to empower new nurses and enhance their organisational commitment and resilience [93] raises questions on the extent of organisational change when power structures remain unaltered. While the nurses may have felt empowered and committed to their organisation's goals, they may not become resilient to the adversities that emanate from their organisation's hierarchies. Ramsey argues that shifting organisational culture to reduce mistreatment will "require strong leadership that affirms emotion work as an explicitly valued task, directing resources and reshaping relationships to affirm trust and respect. This includes ensuring a physically and psychologically safe environment for providers and actively reducing stress and promoting mental well-being" [99].

Still, the intervention studies reviewed highlight several approaches to strengthening relationships between providers and with management that have the potential to improve care provision and buffer the negative effects of shortages. It is also worth noting that the majority of interventions we reviewed focused on training providers and managers. And those that focussed on leadership and supervision included some component of 'group problem solving'. Both these approaches were strongly highlighted in a 2018 review of strategies to improve health worker performance in LMICs, which ultimately recommended combining training with other strategies, such as supervision or group problem solving to increase effectiveness [101].

Ultimately, changes in 'software' elements of organisational management must be matched by serious attempts to address the challenges posed by resource shortages, and with wider improvements in training, renumeration and formal professional recognition that are also powerful levers to tackle medical hierarchies. Nurses, midwives, interns, and early residents provide the majority of intrapartum care for low-risk women giving birth in facilities in LMICs, both by design and necessity. Progress towards RMC hinges on greater attention to their management and welfare.

The review points to some key areas of future research. The majority of articles drawn on to identify organisational problems were qualitative in nature, elicited provider views, and did not set out to capture mistreatment. On the whole resource shortages were linked more directly with provider behaviours and care that constitute mistreatment. While issues

stemming from medical hierarchies still impacted providers and clinical practices negatively, their association with behaviours was less explicit. More research specifically designed to capture organisational issues and test the strength of their associations with mistreatment is needed to adequately inform solutions.

Though we know that mistreatment can be patterned along various socio-economic lines, impacting marginalised groups more severely [3] such findings did not strongly emerge from this review. Also, we acknowledge that there may be individual variation among providers and how they respond to the same stressful working conditions. Future research should aim to account for and explain individual factors (both patient and provider) that may moderate mistreatment. Mixed-methods and other approaches that support triangulation of facility-level data, provider and patient information, are likely to be useful moving forward. Finally, and as for solutions, this review sought to identify organisational issues and interventions in particular. Multipronged approaches that recognise challenges in working conditions along with other drivers of mistreatment are needed to effectively address mistreatment in childbirth and promote RMC.

## Limitations and strengths of the review

While we were able to capture a wide range of organisational issues, our study findings may still be limited, owing to the complexity of the topic and its extensive scope of literature. Language restrictions might have excluded relevant studies from non-English journals (for example, obstetric violence literature from Latin American countries). Additionally, despite covering several databases, our search yielded few intervention studies within LMIC focussed exclusively on maternity care and/or maternity providers. This may be because we restricted our search to peer-reviewed studies, which may have excluded relevant interventions in the non-empirical literature such as national reports, dissertations, etc. This reduced our ability to draw direct linkages from interventions in other medical disciplines to improvements in maternity settings specifically. Conversely, we were unable to assess whether there may be aspects to maternity care that might modify expected results of interventions drawn from other medical disciplines. This is clearly an area that requires further empirical work.

Despite these limitations, to our knowledge this is the first review to focus investigation on organisational issues at the facility-level that impact maternity providers and care across LMIC contexts. Second, given the strong emphasis on resource shortages as a primary barrier to poor care in public institutions, the review raises questions on its relative influence, pointing areas of organisational management that require attention when designing RMC interventions.

## Conclusion

Our review draws attention to several organisational challenges, related to and beyond shortages, that negatively impact providers and their ability and willingness to care for labouring women. Rationalising workloads across cadres, increasing access to training for nurses and midwives, and reducing inter cadre differences in reward systems would contribute to addressing institutional hierarchies that are central to these challenges. Promising interventions have focussed on building the capacity of managers and leaders to institute processes and foster a culture of open communication, knowledge-sharing, support, and trust, which reduce the effects of power inequalities. It is clear that a combination of structural and normative changes that improve organisational conditions for frontline providers are essential for reducing mistreatment and enabling RMC. Further, contextualising such strategies by conducting reflexive implementation research that is grounded within health systems realities is critical to making change sustainable.

## Supporting information

**S1 Table. PRISMA checklist.**
(DOCX)

**S2 Table. Non-intervention studies.**
(DOCX)

**S3 Table. Intervention studies.**
(DOCX)

**S1 Text. Search terms.**
(DOCX)

## Acknowledgments

We thank all members of the WHO working group on interventions to reduce mistreatment of women during childbirth, for their thoughtful comments and suggestions through the process of this study.

## Author Contributions

**Conceptualization:** Bhavya Reddy, Sophia Thomas, Aditi Iyer, Gita Sen.

**Data curation:** Bhavya Reddy, Sophia Thomas, Baneen Karachiwala, Ravi Sadhu.

**Formal analysis:** Bhavya Reddy, Sophia Thomas, Baneen Karachiwala, Ravi Sadhu, Aditi Iyer, Gita Sen.

**Funding acquisition:** Özge Tunçalp.

**Investigation:** Bhavya Reddy, Sophia Thomas, Baneen Karachiwala, Ravi Sadhu.

**Methodology:** Bhavya Reddy, Sophia Thomas, Ravi Sadhu, Aditi Iyer, Gita Sen.

**Project administration:** Bhavya Reddy.

**Supervision:** Aditi Iyer, Gita Sen.

**Validation:** Bhavya Reddy, Sophia Thomas, Baneen Karachiwala, Ravi Sadhu.

**Visualization:** Bhavya Reddy, Sophia Thomas, Baneen Karachiwala.

**Writing – original draft:** Bhavya Reddy, Sophia Thomas, Aditi Iyer, Gita Sen.

**Writing – review & editing:** Bhavya Reddy, Sophia Thomas, Baneen Karachiwala, Ravi Sadhu, Aditi Iyer, Gita Sen, Hedieh Mehrtash, Özge Tunçalp.

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
