## [Decision Letter · Decision Letter 0]

27 Jul 2022

PGPH-D-22-00889

A scoping review of the impact of organisational factors on providers and related interventions in LMICs: Implications for Respectful Maternity Care

Dear Dr. Reddy,

Thank you for submitting your manuscript to PLOS Global Public Health. After careful consideration, we feel that it has merit but does not fully meet PLOS Global Public Health’s publication criteria as it currently stands. Therefore, we invite you to submit a revised version of the manuscript that addresses the points raised during the review process.

Please give careful consideration to the constructive feedback provided by both reviewers below, particularly with regards to choice of phrasing used to describe the study focus, and contextualization of findings both within maternity care and national health systems more broadly.

We look forward to receiving your revised manuscript.

Kind regards,

Hannah Tappis, DrPH, MPH

Academic Editor

Journal Requirements:

1. Please amend your detailed online Financial Disclosure statement. This is published with the article. It must therefore be completed in full sentences and contain the exact wording you wish to be published.

State what role the funders took in the study. If the funders had no role in your study, please state: “The funders had no role in study design, data collection and analysis, decision to publish, or preparation of the manuscript.”

2. Please update your online Competing Interests statement. If you have no competing interests to declare, please state: “The authors have declared that no competing interests exist.”

3. Please provide a complete Data Availability Statement in the submission form. If your research concerns only data provided within your submission, please write “All data are in the manuscript and supporting information files.” as your Data Availability Statement.

4. Please provide separate figure files in .tif or .eps format and remove any figures embedded in your manuscript file. Please also ensure that all files are under our size limit of 10MB.

5. Please ensure that you refer to your figures in your text as, if accepted, production will need these references to link the reader to the figures.

6. We noticed that you have two Figure 1's in your manuscript. Please update your figure numbers and cite them accordingly.

7. Tables cannot contain images. Please remake any tables with images as main figures and provide them as separate one page .tif or .eps files. Please change any in-text citations as necessary.

8. We have noticed that you have uploaded Supporting Information files, but you have not included a list of legends. Please add a full list of legends for your Supporting Information files after the references list. 

Additional Editor Comments (if provided):

Reviewers' comments:

Reviewer's Responses to Questions

**Comments to the Author**

1. Does this manuscript meet PLOS Global Public Health’s publication criteria? Is the manuscript technically sound, and do the data support the conclusions? The manuscript must describe methodologically and ethically rigorous research with conclusions that are appropriately drawn based on the data presented.

Reviewer #1: Partly

Reviewer #2: Yes

2. Has the statistical analysis been performed appropriately and rigorously?

Reviewer #1: N/A

Reviewer #2: N/A

3. Have the authors made all data underlying the findings in their manuscript fully available (please refer to the Data Availability Statement at the start of the manuscript PDF file)?

Reviewer #1: Yes

Reviewer #2: Yes

4. Is the manuscript presented in an intelligible fashion and written in standard English?

Reviewer #1: Yes

Reviewer #2: Yes

5. Review Comments to the Author

Reviewer #1: This is a very interesting and extensive narrative review of a topic of great interest, with implications for future research and evidence-generation to change practice.

My review comments are presented in the annotated version of the manuscript draft that is attached, and as such are not detailed here.

Here, I summarize three overarching issues:

1) The choice to dichotomize by country income levels should be examined and substantiated. There is arguably more generalizability across the range of country income levels with respect to maternal health medical culture, organizational issues, and RMC, than there is across health disciplines. D&A is known to be a universal phenomenon whose manifestations and their magnitude may vary by context. As a form of gender-based violence the link to women's sexual and reproductive health is not incidental. The rationale and support for the extension of the search criteria to include studies from other medical disciplines should be explicitly shared.

2) RMC is not merely the absence of D&A or mistreatment and therefore these terms cannot be used interchangeably. This study seems to explore the association of organizational challenges with the incidence of D&A, so this should be stated.

3) The strength of the evidence and the associations derived from the narrative review of qualitative and intervention studies should be discussed. Overall, it appears that this review provides a strong basis for hypothesis genration and calls for further research to explore the associations and their strength between the organizational issues (independent variable) and their effect on D&A (dependent variable).

There are a number of language issues that could be examined and addressed, such as whether there is a significant distinction between "organizational" and "institutional" and whether "irrational" and "rational" are commonly defined and understood as they are used in this paper. Professional copyediting would strengthen the writing.

Reviewer #2: This is a well written article on an interesting and important topic, and I believe that it should be published.

I found myself thinking about a few issues having read the article that I feel would be helpful to address/discuss. I am not sure if the issue is about the information extracted from the articles –or the articles itself – and I think that it might be the later.

I think when we are researching issues such as this, we must recognise that there are always going to be dominant discourses, it’s hard to explain why something happens -and it’s so easy to blame being for example being overstaffed – and we have to develop research methods / approaches/ questions that go beyond the obvious.

In a former life – not as an academic but a government official – I found when visiting facilities that everyone said that they were overworked – but there was a huge variation of workloads. Some maternity wards were totally overloaded – some were overloaded some of the time – and some were not. Similarly, I found variation in the care that was given by staff – with some being nicer to patients than others – despite the challenging circumstances. I have always been frustrated that our methods and approaches/ reporting of work in this to issue often don’t give space for individual variation. This not to say there are not structural issues – but I think there are also individual variations – and we need to think about how to support good providers!

In the background (line 9/10) the author states that “we have significant evidence that women, especially those who are disadvantaged and marginalised, experience forms of mistreatment”. In the results reported from the review there did not seem to be information reported about why organisational factors might lead to some women being treated worse than others – explanations were just given why overall treatment might be poor quality. Is this because it was not explored in the literature? I think it would be worth reflecting on.

I had similar questions about ‘structural gender inequality’ and ‘women’s low status in the community’. Did any of the articles address these issues?

Another question that I had – and maybe worth at least addressing in the discussion is around the specifics of disrespect and abuse in maternity services. I found myself thinking when I read the article – that workload/ poor supervision/ organisational dynamics – don’t seem that particular to maternity services – but there do seem to be some specific problems that exist in maternity services.

So, in conclusion – this is a great article – but I found myself wanting a better link between the intro and discussion – and either extraction of data related to social inequalities – or recognition that this is not being addressed.

6. PLOS authors have the option to publish the peer review history of their article (what does this mean?). If published, this will include your full peer review and any attached files.

**Do you want your identity to be public for this peer review?** For information about this choice, including consent withdrawal, please see our Privacy Policy.

Reviewer #1: No

Reviewer #2: No

---

## [Editor Report · Decision Letter 1]

15 Sep 2022

A scoping review of the impact of organisational factors on providers and related interventions in LMICs: Implications for Respectful Maternity Care

PGPH-D-22-00889R1

Dear Ms. Reddy,

We are pleased to inform you that your manuscript 'A scoping review of the impact of organisational factors on providers and related interventions in LMICs: Implications for Respectful Maternity Care' has been provisionally accepted for publication in PLOS Global Public Health.

Best regards,

Hannah Tappis, DrPH, MPH

Academic Editor